

# Retrieval of intrinsic mesospheric gravity wave parameters using lidar and airglow temperature and meteor radar wind data

Robert Reichert[1], Bernd Kaifler[1], Natalie Kaifler[1], Markus Rapp[1,2], Pierre-Dominique Pautet[3], Michael J. Taylor[3], Alexander Kozlovsky[4], Mark Lester[5], and Rigel Kivi[6]

[1]Deutsches Zentrum für Luft- und Raumfahrt, Institut für Physik der Atmosphäre, Oberpfaffenhofen, Germany
[2]Meteorologisches Institut, Ludwig-Maximilians-Universität, Munich, Germany
[3]Utah State University, Logan, USA
[4]Sodankylä Geophysical Observatory, Finland
[5]University Leicester, United Kingdom
[6]Space and Earth Observation Centre, Finnish Meteorological Institute, Sodankylä, Finland

*Correspondence to:* Robert Reichert (robert.reichert@dlr.de)

**Abstract.** We analyze gravity waves in the mesosphere, lower thermosphere region from high-resolution temperature variations measured by Rayleigh lidar and OH temperature mapper. From this combination of instruments, aided by meteor radar wind data, the full set of ground-relative and intrinsic gravity wave parameters are derived by means of the novel WAPITI method. This Wavelet Analysis and Phase line IdenTIfication tool decomposes the gravity wave field into its spectral com-

ponent while preserving the temporal resolution, allowing us to identify and study the evolution of gravity wave packets in the varying background. We describe WAPITI and demonstrate its capabilities for the large-amplitude gravity wave event on 16/17 December 2015 observed at Sodankylä, Finland, during the GW-LCYCLE-II field campaign. We present horizontal and vertical wavelengths, phase velocities, propagation directions and intrinsic periods including uncertainties. The results are discussed for three main spectral regions, representing short, medium and long-period gravity waves. We observe a complex

superposition of gravity waves at different scales, partly generated by gravity wave breaking, evolving in accordance with a vertically and presumably also horizontally sheared wind.

## 1 Introduction

The impact of atmospheric gravity waves (GW) on the energy and momentum budget especially in the upper mesosphere, lower thermosphere (MLT) region has long been recognized (Lindzen, 1981; Holton, 1982, 1983; Vincent and Reid, 1983).

Yet, many mechanisms are not fully understood today, for example regarding generation, intermittency, interactions or breaking of GWs (see Fritts and Alexander, 2003, for a review). Understanding of these processes requires detailed case studies with a complete description of intrinsic gravity wave parameters and the atmospheric background. A well-established and capable technique to observe mesospheric GWs is the imaging of OH layer emissions, which provides compelling detail of the spatial and temporal characteristics of GWs (Swenson and Mende, 1994; Taylor et al., 1995; Nakamura et al., 2003; Suzuki

et al., 2007; Vargas et al., 2016). Analysis methods applied to images of the OH layer in order to infer horizontal wavelengths, ground-relative wave periods and phase speeds as well as propagation directions include filtering with selected bandwidths





e.g. to enhance GW signatures, fitting routines to specific, e.g. cyclic, wave structures (e.g. Hapgood and Taylor, 1982), time-difference and correlation techniques (Swenson et al., 1999; Tang et al., 2003), cross-spectral and wavelet analysis (Frey et al., 2000), three-dimensional fast Fourier transform (Matsuda et al., 2014), and maximum entropy methods (Sedlak et al., 2016). In order to retrieve horizontal wavelengths larger than the FOV of the imager, Takahashi et al. (2009); Fritts et al. (2014)

analysed keogram representations of airglow imager data. In addition to process studies, also long-term observations of OH layer emissions with imaging instruments are employed to derive statistics and seasonal variations of GW parameters at different locations (Walterscheid et al., 1999; Nakamura et al., 1999; Suzuki et al., 2004; Li et al., 2016, 2018; Shiokawa et al., 2009).

Limitations of those techniques are the variable height and width of the OH layer with impact on the display of GWs de-
pending on their periods and vertical wavelengths (Gardner and Taylor, 1998; Dunker, 2018). To obtain intrinsic periods, OH imaging data is often combined with meteor radar observations of the ambient wind (e.g. Nyassor et al., 2018). Mangognia et al. (2016) proposed a multi-channel instrument to deduce vertical wavelengths which must otherwise be determined from the dispersion relation or in combination with other observation techniques. Especially sodium lidar instruments, providing vertical soundings of vertical wind and temperature in the sodium layer located above the OH layer, were essential in order to
derive vertical wavelengths, phase speeds and momentum fluxes in the MLT region (Swenson et al., 1999; Jia et al., 2016). This technique allowed for the study of GW dispersion, refraction or breaking in the MLT region (Smith et al., 2005; Yuan et al., 2016). Isler et al. (1997) and Hecht et al. (1997) also incorporated wind measurements by MF radar to investigate ducting, evanescence and breaking of GWs.

Lidar soundings of the middle atmosphere are ideally suited to study gravity waves at high vertical and temporal resolution (e.g. Hostetler and Gardner, 1994; Lu et al., 2009; Baumgarten et al., 2015; Zhao et al., 2017; Fritts et al., 2018). In the stratosphere and lower mesosphere, climatologies of gravity wave potential energy densities were derived from lidar backscatter (Wilson et al., 1991; Sivakumar et al., 2006; Sica and Argall, 2007; Thurairajah et al., 2010; Alexander et al., 2011; Mzé et al., 2014; Kaifler et al., 2015; Kogure et al., 2018). Due to limitations in power and/or efficiency of Rayleigh lidars, often
a gap remains in the upper mesosphere between the top altitude of Rayleigh lidar temperature profiles and coincident sodium lidar measurements. Recent developments in Rayleigh lidar technology allow for high-resolution temperature and GW measurements at altitudes above ∼85 km within the OH layer (Kaifler et al., 2017, 2018). Also, modern OH imaging instruments utilize different OH emission lines to perform spectroscopy in order to derive the spatial temperature field, thus facilitating a synergy of OH layer imaging and vertical lidar soundings that has not been possible before.

In recognition of the prospects of multi-instrument approaches for middle atmosphere GW studies, several field campaigns combining a large number of ground-based and air-borne instruments have been undertaken in recent years, e.g. the GW-LCYCLE-1 campaign in northern Scandinavia in 2013 (Wagner et al., 2017; Witschas et al., 2017) and the DEEPWAVE campaign in New Zealand in 2014 (Fritts et al., 2016). During the winter of 2015/16, the season when GWs of high amplitude
are able to propagate from the troposphere to the mesosphere, the GW-LCYCLE-II aircraft campaign took place in northern



Scandinavia. During and beyond the campaign period, the high-power CORAL Rayleigh lidar, the AMTM OH imager and the SKiYMET meteor radar were co-located at Sodankylä, Finland. For the first time, common-volume observations at resolutions below one hour can thus be exploited for detailed studies of gravity waves in the MLT region. We develop a consistent analysis method to derive the full set of GW parameters by means of wavelet analysis and a phase line identification (WAPITI) algo-

rithm and apply it to the combined data set. Wavelet analysis is a well established method in atmospheric science in general, but also with respect to the study of atmospheric GWs. It is applied to in-situ, remote-sensing and satellite observations as well as reanalysis data with the goal of characterizing GW parameters, derive global maps or study generation processes in the troposphere or stratosphere and turbulence generation by breaking GWs (Stockwell et al., 1996; Alexander and Barnet, 2007; Zhang et al., 2001; Dörnbrack et al., 2018; Koch et al., 2005). The key advantage is the preserved temporal resolution,

allowing for the detection of spatially and temporally localized wave packets. We present a systematic spectral decomposition of AMTM keograms (a two-dimensional representation of pixel columns and rows evolving with time, see e.g. Taylor et al., 2009) and lidar data in order to detect and characterize GW packets observed by the three instruments. Using this technique, co-existing, interacting and evolving GW packets can be studied.

We demonstrate our analysis for the GW event on 16/17 December 2015, when large-amplitude propagating GWs were observed in the mesosphere. The instruments are described in Sect. 2 together with their respective data sets. The WAPITI algorithm developed to derive GW parameters like horizontal and vertical wavelengths, propagation directions and phase velocities for waves of different scale is described in Sect. 3. It is applied to the data sets of 16/17 December 2015 in Sect. 4, followed by a discussion of the spectral characteristics of the identified waves in Sect. 5. Conclusions are given in Sect. 6.



## 2 Instruments and data sets

During the GW-LCYCLE-II (Gravity Wave Life Cycle Experiment) campaign in winter 2015/16, three instruments were co-located at Sodankylä, Finland (67.4° N, 26.6° E), collecting complementary data sets. The Compact Rayleigh Autonomous Lidar (CORAL) provided vertical profiles of middle atmospheric temperature and GW perturbations (27 – 98 km). The Advanced Mesospheric Temperature Mapper (AMTM) detected GWs in the horizontal plane at the altitude of the OH layer (∼ 86 km). The Sodankylä-Leicester Ionospheric Coupling Experiment (SLICE) meteor radar provided horizontal wind measurements in the upper mesosphere (82 – 98 km). Simultaneous observations of all three instruments were obtained during 78 nights between September 2015 and April 2016. Here we selected the night of 16/17 December 2015 when large-amplitude GWs occurred in order to demonstrate our retrieval of the full set of GW parameters.

### 2.1 CORAL

CORAL was built by the German Aerospace Center (DLR) and installed at the Finnish Meteorological Institute Sodankylä site in September 2015. CORAL is a Rayleigh backscatter lidar for the middle atmosphere. It incorporates a 12 W laser operated at 532 nm wavelength as transmitter, a 63 cm diameter receiving telescope and two height-cascaded receiving channels equipped with Avalanche Photo Diodes run in photon-counting mode (Kaifler et al., 2017, 2018). CORAL was designed with the capability for remote control and automatic operation in order to maximize operation hours. During six months of operation, 492 hours of high-quality data were collected during night-time. In the absence of aerosols, the received photon counts are directly proportional to the density of the atmosphere. Following Hauchecorne and Chanin (1980) we infer atmospheric temperatures by top-down integration of atmospheric density profiles assuming hydrostatic equilibrium. This is first performed for the nightly average profile smoothed with a filter of ∼2 km width using SABER measurements as a seed value at 100 – 110 km altitude. Then, in an iterative manner, temperature profiles of subsequently higher resolution of 120 min, 60 min, 30 min, 20 min and 10 min are obtained by seeding at their top altitudes with the respective lower-resolution profiles. For this study, we use temperature data at 2070 m vertical and 10 min time resolution computed on a grid of 90 m × 1 min which is intrinsic to the temperature retrieval. Above ∼92 km the temporal resolution gradually decreases from 10 min to 2 h due to longer integration times. The uncertainties in absolute temperatures caused by photon noise and in part by the temperature seeding process, amount to 0.3 K between 25 – 50 km, 1.6 K between 50 – 75 km and 9.9 K between 75 – 100 km. Figure 1 shows the temperature measurement of 16/17 December 2015. We see the stratopause at 60 km with strong wave activity above in the mesosphere with periods of about 4 h. A more detailed description is given in Sect. 3.



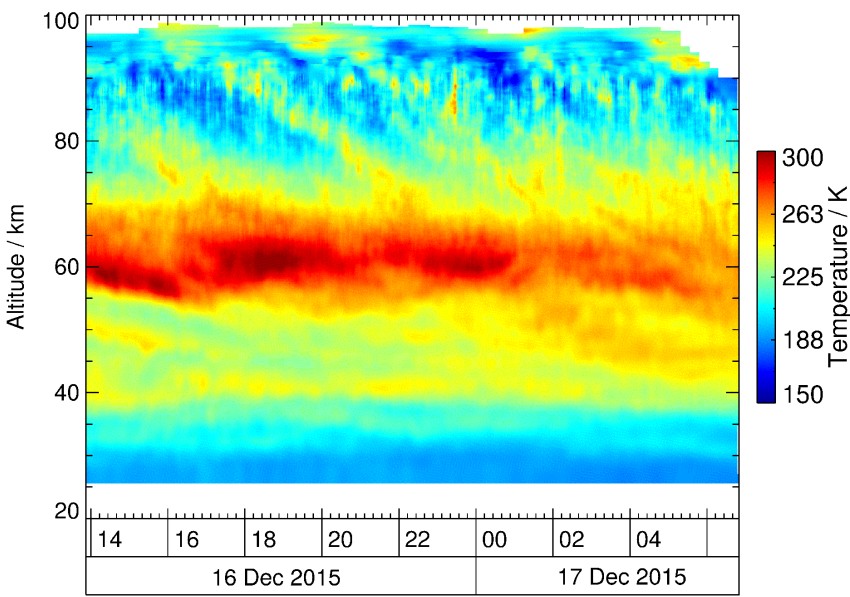

**Figure 1.** Time-altitude section of temperature obtained by CORAL on 16/17 December 2015. Above ∼92 km, the temporal resolution decreases gradually from 10 min to 2 h.

## 2.2 AMTM

The AMTM was built by Utah State University and yields temperature maps based on detected IR radiation originating from the OH layer (Pautet et al., 2014; Fritts et al., 2014). This layer is commonly described as Gaussian shaped with an average peak altitude of 86.8±2.6 km and a FWHM of 8.6±3.1 km, although the local height and thickness can vary (Baker and Stair Jr,

5    1988). The temperature is a function of the brightness ratio of two spectral lines in the Meinel (3,1) rotation-vibration hydroxyl bands namely, $B[P_1(2)]/B[P_1(4)]$. The spatial and temporal resolution of the AMTM is 625 m at zenith and ∼30 s, and the field of view is about 200 km × 160 km. Temperature uncertainties are in the order of $1-2$ K for clear sky conditions. Due to strong contamination by sunlight, the AMTM operates only during darkness. To gain better insight into the temporal evolution of the temperature maps we analyse AMTM data in the keogram representation. The AMTM's FOV is oriented in the cardinal

10    directions as shown in Fig. 2a. Concatenating pixel rows (columns) of successive temperature maps at ∼30 s resolution results in a South-North (West-East) keogram projecting GWs onto this particular direction. AMTM data for 16/17 December 2015 in this representation are shown in Fig. 2b. The position of the lidar laser beam is in the zenith of the AMTM's FOV as indicated by the dashed lines in Fig. 2b. For the analysis we interpolated the AMTM data sets on a one-minute grid to be able to compare it with lidar data and to reduce noise.


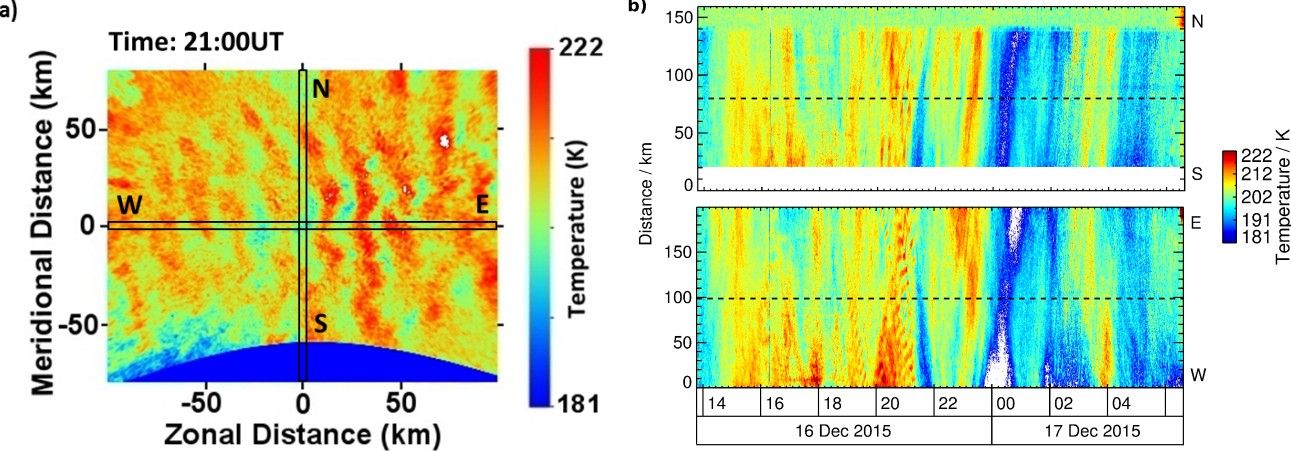

**Figure 2.** (a) AMTM temperature map at 21 UT on 16 December 2015 with indicated pixel rows and columns used for keogram representation. (b) South-North keogram of AMTM temperature measurements (top) and West-East keogram (bottom) on 16/17 December 2015. The central dashed lines indicate the position of the lidar laser beam.

## 2.3 Meteor radar

SLICE is a SKiYMET meteor radar (MR) and provides measurements of horizontal wind speeds in the altitude range 82 – 98 km at a vertical and temporal resolution of 2.7 km × 1 h (Lukianova et al., 2018). The MR is operated by the Sodankylä Geophysical Observatory. Wind data are retrieved fom the line-of-sight velocity of the ionized meteor trails detected in a FOV of about 300 km in diameter (Hocking et al., 2001). Figure 3a-c show zonal and meridional wind speeds as well as the wind direction during the night of 16/17 December 2015. The wind field is complex with both vertical and horizontal reversals and shears on short scales. We therefore expect time-varying parameters of GWs observed by CORAL and the AMTM.





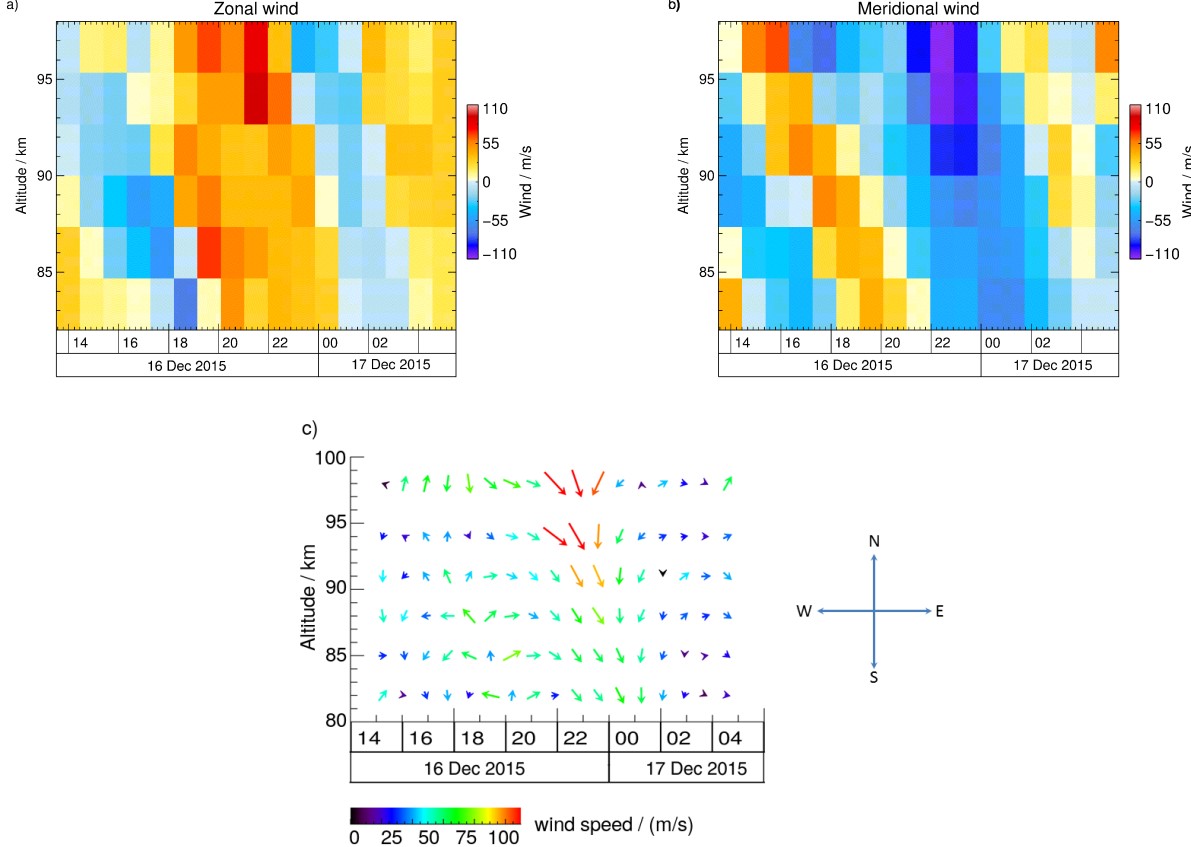

**Figure 3.** (a) Zonal wind, (b) meridional wind, and (c) wind direction observed by the SLICE meteor radar on 16/17 December 2015.





## 3 Analysis

In the following, we present our newly developed Wavelet Analysis and Phase line IdenTIfication (WAPITI) algorithm used to retrieve observed and intrinsic GW parameters based on three complementary data sets at mesospheric altitudes. From AMTM temperature data we obtain horizontal wavelengths as well as propagation directions at the altitude of the OH layer.

Vertical wavelengths and directionality of the vertical propagation are derived from CORAL vertical temperature profiles. In combination with meteor radar wind speeds we estimate intrinsic periods. As a crosscheck, we use the dispersion relation to retrieve vertical wavelengths based on horizontal wavelengths and intrinsic periods and compare the result with vertical wavelengths retrieved from lidar data.

### 3.1 Spectral filtering of temperature data

The AMTM provides data in temporally resolved keogram representation in horizontal dimensions $x$ and $y$, while CORAL observations yield time series of temperature at different altitudes $z$. Following Torrence and Compo (1998) we apply a wavelet transformation with sixth-order Morlet wavelets to all the time series within the common FOV of both instruments, i.e. for the lidar position at OH layer altitudes.

The discrete periods used for the wavelet transformations are given by

$$\tau_j = \frac{4\pi \cdot dt \cdot 2^{j/8+1}}{\omega_0 + \sqrt{2 + \omega_0^2}}. \tag{1}$$

Here, the non-dimensional frequency $\omega_0 = 6$, which is the order of the wavelet. The time resolution $dt = 1\,\mathrm{min}$, and $j$ is the index of the wavelet ranging from $0 - 71$, selecting periods between $2\,\mathrm{min}$ and $16.2\,\mathrm{h}$. The resulting wavelet spectra show spectral power based on squared amplitudes of temperature perturbations as a function of time and observed period (Fig. 4). The cone of influence (COI), inside which edge effects due to the finite time series may result in an underestimation of spectral

power, is marked as hatched region. Significance levels of 95%, 50% and 20% calculated relative to red noise spectra are shown as contour lines to highlight potential GW structures within each spectrum. To assess the effect of temperature uncertainties on the wavelet spectrum we perform Monte Carlo simulations. We add 100 times Gaussian-distributed white noise with a standard deviation of the temperature measurement uncertainty to the actual temperature time series and apply a wavelet transformation. The average standard deviation is shown as dashed line together with the global wavelet spectra (Fig. 4bce).

In order to compare wavelet spectra of CORAL and the AMTM we weighted lidar temperatures between $78 - 95\,\mathrm{km}$ with a Gaussian with FWHM of $8.6\,\mathrm{km}$ centered at $86.8\,\mathrm{km}$. Figure 4a shows the natural logarithm of the squared absolute value of the spectral amplitude of the CORAL temperature time series averaged over the OH layer. An impression of the spectral variability with height is given in Fig. 4c where we present global wavelet spectra at different altitudes. The Monte Carlo simulation revealed increasing noise levels in global wavelet spectra with smaller periods especially below $0.5\,\mathrm{h}$. Hence, we

decided to focus on the spectral range between $0.5\,\mathrm{h}$ and $8\,\mathrm{h}$ observed period. Figure 4d displays the wavelet spectrum for the zenith time series of AMTM temperature (dashed lines in Fig. 2b). A detailed description of the spectra is given in Sect. 4.



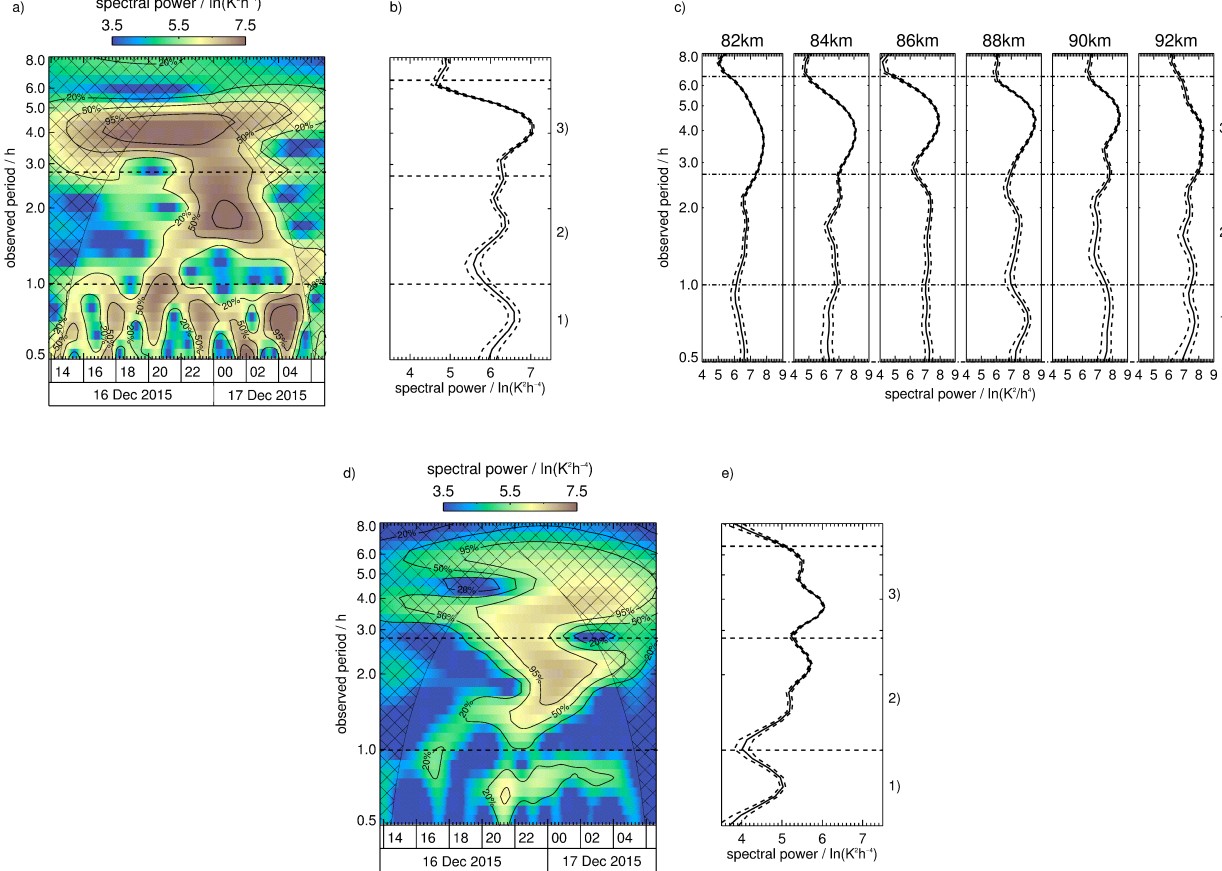

**Figure 4.** (a) Wavelet power spectrum of Gaussian weighted lidar time series and (b) global wavelet spectrum including standard deviation. (c) CORAL global wavelet spectra including standard deviations for six altitudes between $82 - 92$ km. (d) Wavelet power spectrum of the zenith time series of the AMTM and (e) global wavelet spectrum including standard deviation. The hatched areas mark the COI. Solid lines indicate the 20%, 50% and 95% significance levels. Three period ranges with dominant waves are indicated by numbers on the right axis.

We analyse wavelet power at 32 discrete periods $\tau_j > 30$ min and reconstruct the respective temperature perturbations for each spatial dimension $x$, $y$ (AMTM) and $z$ (CORAL). If GWs with periods $\tau_j$ are present in the data set, they are observed as a pattern of phase lines in the wavelet reconstruction. Two examples are shown in Fig. 5bc. Periods not supported by the data exhibit very small reconstructed temperature perturbations. From the progression of phase lines we derive phase velocities in each direction.

## 3.2 Phase line identification

In order to detect phase lines in the reconstructed data, we need to identify points belonging to the same phase, e.g. points along wave crests or troughs. They are detected by looking for a change of sign in the derivative of the temperature time series





with respect to time. In order to allow for a robust phase line detection, the wavelet reconstructions are smoothed with a boxcar window with a width of 5 km in the spatial domain for keograms and 2 km for vertical profiles, and $\tau_j/4$ in the time domain. The identified extrema are connected in space in order to determine the phase angle in the zenith position by a linear fit to phase lines.

**Figure 5.** (a) Sketch illustrating the phase line detection of the WAPITI algorithm. The grid represents data resolved in space and time. Grey boxes represent local temperature maxima in each time series $r_i$. Blue arrows point to adjacent maxima that are affiliated with a phase line. Red arrows point to adjacent maxima that are too far away and therefore do not belong to the considered phase line. Solid arrows describe the first step in the algorithm and sketched time intervals refer to these solid arrows. Dotted arrows describe further steps of the algorithm. (b) Phase line identification applied to reconstructed temperature perturbations in lidar data with a period of 4.0 h and (c) in airglow data with a period of 0.7 h. Solid black lines are fitted linear functions.





Figure 5a illustrates the phase line detection of our WAPITI algorithm. To isolate a number of $i$ phase lines, we find points of time $t_i(r)$ which follow the position of wave crests and troughs in each data set. For example, the first maximum in the zenith time series occurs at $t_1(r_0)$. In the time series adjacent to $r_0$ we find the maximum which is closest to $t_1$ and at most $\Delta t < \tau_j/2$ apart. This maximum is identified as belonging to the phase line $i = 1$. This step is repeated for all $r$ within the

window $\Delta r = 12\,\frac{\text{km}}{\text{h}}\tau_j + 20\,\text{km}$ for keograms and $\Delta r = 8.6\,\text{km}$ for the lidar data set. The window width for lidar data contains the average thickness of the OH layer. We choose a dynamic range for keograms, i.e. the range increases with the selected period $\tau_j$, because larger structures tend to be more coherent. In a second step we fit a linear function $t_i(r) = a_i r + b_i$ to the detected phase lines and calculate the 1-sigma uncertainty estimates $\Delta a_i$ and $\Delta b_i$. The coefficient $a_i$ has the dimension of an inverse velocity, while $b_i$ is a constant time parameter.

Examples of reconstructed time series and detected phase lines are shown in Fig. 5bc. Increased amplitudes indicate the presence of GWs with the selected $\tau_j$. The matching of temperature reconstructions and estimated phase lines is very good for short as well as for large periods.

### 3.3 Horizontal wavelength and direction of propagation

The observed horizontal wavelength $\lambda_h$ can be retrieved directly from the OH airglow images. However, in this case, the

maximum wavelength is in the order of the dimension of the FOV ($\sim$200 km). Larger horizontal wavelengths can be retrieved from the detected phase lines from above. Then, $\lambda_h$ is given by the product of the phase velocity $c$ and the period of the GW $\tau$. Hence, in order to retrieve the phase velocities $c_i$ in each direction at time $t_i(r_0)$, we calculate the inverse of the derivative of the linear fit $t_i(r)$ with respect to space,

$$c_i(t_i(r_0)) = \left(\frac{dt_i(r)}{dr}\right)^{-1} = \frac{1}{a_i}. \tag{2}$$

To estimate the uncertainty of the phase velocity $\Delta c_i$ we calculate the propagated error of $\Delta a_i$ given by

$$\Delta c_i(t_i(r_0)) = c_i(t_i(r_0))^2 \Delta a_i. \tag{3}$$

We now have an estimate for phase velocity and its uncertainty at discrete points in time. As an approximation, we assume that the fitted parameters vary linearly between these discrete points. Consequently, $a_i$, $b_i$, $\Delta a_i$ and $\Delta b_i$ are linearly interpolated. To simplify the equations, we drop the explicit time dependence from now on in the notation. We obtain wavelengths $\lambda_x$ and

$\lambda_y$ by

$$\lambda_x = \tau c_x \tag{4}$$

$$\lambda_y = \tau c_y. \tag{5}$$

Uncertainties for these and following quantities are given in appendix A. We consider no uncertainty for the period of the wavelet transformation defined in Eq. (1). We also note that uncertainties in the spectral amplitude of the wavelet spectrum

do not lead to uncertainties in the reconstructed period but in uncertainties of the amplitude of the reconstructed signal. Thus,





structures are conserved.

The horizontal wavelength is determined by $\lambda_x$ and $\lambda_y$ (Eq. (5)) as illustrated in Fig. 6a.

$$\lambda_h = \frac{\lambda_y \lambda_x}{\sqrt{\lambda_y^2 + \lambda_x^2}} \qquad (6)$$

As the phase velocities can have positive and negative signs, $\lambda_h$ can be positive or negative depending on the propagation

direction. However, the direction of propagation is usually expressed as an angle relative to north, thus making the information encoded in the sign of the horizontal wavelength redundant. Following this convention, we drop the sign and consider absolute values only.

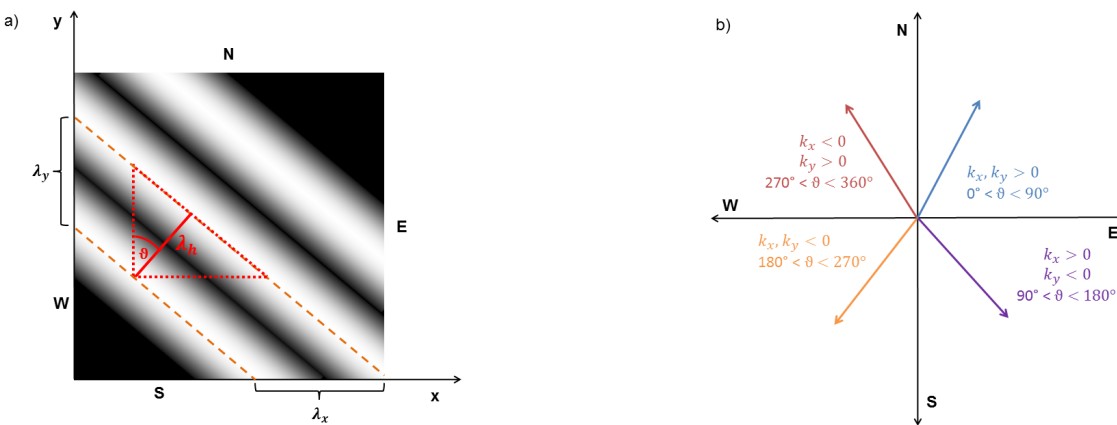

**Figure 6.** (a) Sketch to illustrate the relation between wavelengths. $\lambda_h$ is defined as the height of a right triangle (red) with sides $\lambda_x$ and $\lambda_y$. (b) Definition of the propagation angle. The horizontal direction of propagation is given by the relation of wavelengths in x- and y-direction. Colored arrows describe different propagation directions.

The propagation angle $\theta$ is defined clockwise starting with $\theta = 0°$ being north. Figure 6b illustrates propagation directions in the horizontal plane. For $\lambda_x > 0$ the wave propagates eastward and $\theta = \frac{\pi}{2} + \tan^{-1}\left(-\frac{\lambda_x}{\lambda_y}\right)$. For $\lambda_x < 0$ the wave propagates

westward and $\theta = \frac{3}{2}\pi + \tan^{-1}\left(-\frac{\lambda_x}{\lambda_y}\right)$. With knowledge of wave propagation and wind direction, we calculate the background wind $u_0$ in the direction of propagation, which is needed for the estimation of intrinsic parameters.

### 3.4 Intrinsic period

The observed phase velocity of a GW is given by $c = c_I + u_0$ (Nappo, 2002) with $c_I$ being the intrinsic phase velocity and $u_0$ the background wind. Furthermore, $c_I = \frac{\lambda_h}{\tau_I}$ with $\tau_I$ the intrinsic period and $\lambda_h$ the horizontal wavelength. If we rearrange this

definition and solve for the intrinsic period $\tau_I$, we obtain





$$\tau_I = \frac{\lambda_h}{c - u_0}. \tag{7}$$

The background wind in the direction of propagation is calculated as

$$u_0(\theta) = U\sin(\theta) + V\cos(\theta), \tag{8}$$

with $U$ and $V$ being the zonal and meridional wind components. Both components are weighted with a Gaussian over an
altitude range of $82 - 95\,\mathrm{km}$ with FWHM of $8.6\,\mathrm{km}$ centered at $86.8\,\mathrm{km}$ and averaged over this range. For $u_0 = 0$, the intrin-
sic period equals the observed period. Assuming a constant horizontal wavelength, i.e. uniform horizontal wind (Marks and
Eckermann, 1995), the intrinsic period becomes larger than the observed period if the background wind is oriented in the same
direction as the observed phase velocity and $0 < u_0 < c$. The intrinsic period becomes smaller than the observed period if the
background wind is oriented against the observed phase velocity and $u_0 < 0$. The intrinsic frequency of vertically propagating
GWs is limited to $f < \Omega < N$ (Nappo, 2002), with $N$ the Brunt-Väisälä frequency and $f$ the Coriolis parameter. We estimate
the range for $\tau_I$ to $5\,\mathrm{min} < \tau_I < 13\,\mathrm{h}$ for typical values of $N$ and $f$ at $67.4°\,\mathrm{N}$. For $u_0 = c$ the intrinsic period becomes infinite
and the wave reaches a critical level and breaks. The uncertainty for the intrinsic period is given in Appendix A.

### 3.5 Vertical wavelength

The vertical wavelength $\lambda_z$ is derived using two independent methods. From the phase line detection applied to lidar data
we derive the vertical phase velocity $c_z$. Multiplying it with the period $\tau$ yields the vertical wavelength $\lambda_z$. The uncertainty
$\Delta\lambda_z = \tau\Delta c_z$ is given by the 1-sigma uncertainty estimate from the linear fit applied during the phase line detection. Like the
observed horizontal phase velocity, the observed vertical phase velocity can have both signs as well. For upward propagating
waves $c_z < 0$ and for downward propagating waves $c_z > 0$. According to Dörnbrack et al. (2017) this condition holds for
$u_0 > -c_I$. Otherwise waves appear upward propagating in lidar data while they are in reality downward propagating and vice
versa.

The second approach uses the dispersion relation and the parameters retrieved from AMTM data to derive a vertical wavelength.
In Sect. 4 we show both results for $\lambda_z$ which we discuss in Sect. 5. The dispersion relation reads

$$m^2 = \frac{N^2}{(c - u_0)^2} + \frac{u_0''}{(c - u_0)} - \frac{1}{H_s}\frac{u_0'}{(c - u_0)} - \frac{1}{4H_s^2} - k^2, \tag{9}$$

where m is the vertical wave number and $H_s$ is the scale height (Nappo, 2002). Primes indicate the derivative with respect to
$z$. We substitute $c_I = \lambda_h/\tau_I = c - u_0$ and when we solve the dispersion relation for $\lambda_z$ we get

$$\lambda_z = \frac{2\pi}{\sqrt{\left(\frac{N\tau_I}{\lambda_h}\right)^2 + \frac{u_0''\tau_I}{\lambda_h} - \frac{u_0'\tau_I}{H_s\lambda_h} - \left(\frac{1}{2H_s}\right)^2 - \left(\frac{2\pi}{\lambda_h}\right)^2}}. \tag{10}$$


We derived expressions for $\lambda_h$ and $\tau_I$ in previous sections. $H_s$ is defined as $H_s = RT/g$ with $R = 287\,\mathrm{J\,kg^{-1}\,K^{-1}}$ the universal gas constant for dry air and $g = 9.81\,\mathrm{m\,s^{-2}}$ the acceleration due to gravity. The Brunt-Väisälä frequency $N$ is defined as

$$N = \sqrt{\frac{g}{T}\left(\frac{\partial T}{\partial z} + \frac{g}{c_p}\right)} \tag{11}$$

with $c_p = 1.005\,\mathrm{kJ\,kg^{-1}\,K^{-1}}$ the specific heat capacity for air at constant pressure. We calculate the nightly mean of $N$ in

5    the altitude range $82 - 91\,\mathrm{km}$ based on background temperature profiles which are obtained by low pass filtering of lidar temperature profiles following Ehard et al. (2015). We derive $N = 0.0202 \pm 0.0016\,\mathrm{s^{-1}}$ and $H_s = 5.60 \pm 0.08\,\mathrm{km}$ at OH layer altitudes. We assume $N$ and $H_s$ to be constant and also assume a uniform wind in the horizontal plane, i.e. constant $\lambda_h$. The derivatives of horizontal wind with respect to altitude $u_0'$ and $u_0''$ are determined between $82 - 95\,\mathrm{km}$ and averaged over this altitude range in the same way we did the Gaussian weighted average of the background wind (see Sect. 3.4). In the next

10   section we apply our WAPITI algorithm to GWs observed on the 16/17 December 2015.



## 4 Results

During the night of 16/17 December 2015 strong GW signals were detected by the AMTM and CORAL above Sodankylä, Finland. The wavelet spectra (Fig. 4) reveal a broad distribution of observed periods with high spectral power throughout the night. In lidar data we find three regions with significance levels >95%. The most prominent one lies at ∼4 h observed period

between 17 – 02 UT. Another one lies close to 2 h observed period at 0 UT while the third region consists of three peaks between 0.5 – 1.0 h observed period and 23 – 05 UT (Fig. 4a). In AMTM data we see GW amplitudes with significance levels >95% from 19 UT till the end of the night. Dominant GWs exhibit observed periods in the range 1.5 – 6.5 h (Fig. 4b). We calculate global wavelet spectra (Fig. 4bcd) and divide them into three regions based on local maxima in spectral power. Region 1) is defined from 30 – 60 min observed period and contains short-period GWs. The second region ranges from 1.0 – 2.8 h. Region

3) comprises periods 2.8 – 6.5 h. Please note that region 3) lies to a large extent within the COI and amplitudes might therefore be underestimated. In general, the spectra of AMTM and CORAL are in good agreement. However, amplitudes in the lidar spectrum are larger.

As stated in Sect. 3 we retrieved wavelengths and propagation angles for 32 reconstructed periods such that the GW param-

eters can be displayed as a function of time and observed period. For an overview, the retrieved parameters in the three spectral regions defined above are organised in probability density distributions next to the full spectra. We do a kernel density estimation (KDE) for this purpose. Each value is represented as a normalised Gaussian distribution with a standard deviation given by its uncertainty. Finally all Gaussians are added and divided by the number of values taken into account in order to normalise the probability density distribution. Values with small uncertainties are represented as peaks with small FWHM while values

with large uncertainties are represented as flat and broad peaks in the distribution. In comparison to a standard histogram the KDE takes into account the uncertainty of each value. Hence, the distribution is independent of a chosen bin size and each peak is reliable. In the background we show the total density distribution of each parameter comprising all values within the 20% significance contour line in light grey. This enables us to investigate how each spectral region contributes to the total distribution. Additionally we hatched the parts of each distribution which show parameters within the COI to be aware of their

contribution. Figure 7ab shows horizontal wavelengths retrieved from AMTM data. We see a large variation in $\lambda_h$ ranging from 75 – 2000 km. The uncertainties lie in the order of a few percent in significant regions. Figure 7c shows horizontal propagation directions of the waves identified by the WAPITI algorithm. The density distributions in Fig. 7d show predominantly waves propagating in northward directions. Figure 8ab displays the wind every wave is exposed to, i.e. wind speed in the direction of propagation. Most waves are propagating against the mean flow resulting in negative wind speeds of up to -80 m/s. Figure

8cd shows estimated intrinsic periods based on wind speeds. Due to the motion in opposite directions the majority of intrinsic periods is Doppler-shifted to larger observed periods. Vertical wavelengths retrieved from lidar data are shown in Fig. 9ab where we find values of 6 – 50 km. A background wind field varying in space and time has an impact on the observed period of GWs and hence affects the temperature reconstructions. Therefore phase lines in our data sets are bent and we find locally large wavelengths. Vertical wavelengths between 1 – 50 km are retrieved using the dispersion relation (Fig. 9cd).



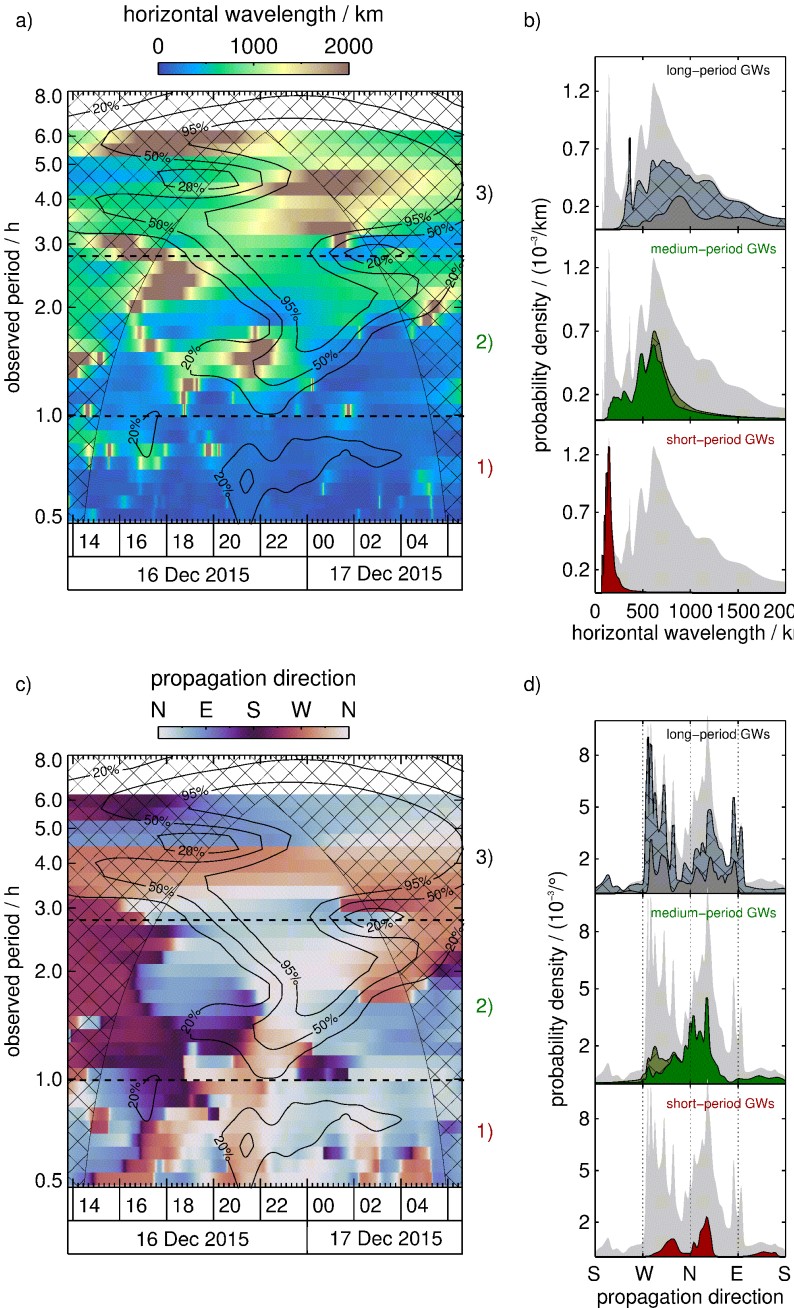

**Figure 7.** (a) Horizontal wavelengths observed by the AMTM as function of time and observed period. The hatched area represents the COI. Contour lines mark the 20%, 50% and 95% significance levels. (b) Density distributions for significance levels >20% for region 1 (red), region 2 (green) and region 3 (grey). The grey distribution in the background shows the total probability density distribution of all values with significance level >20%. Hatched areas indicate values within the COI. (c) Same as (a) for propagation directions based on horizontal wavelengths. (d) Density distributions for each region as in (b). Dotted lines mark the cardinal directions.



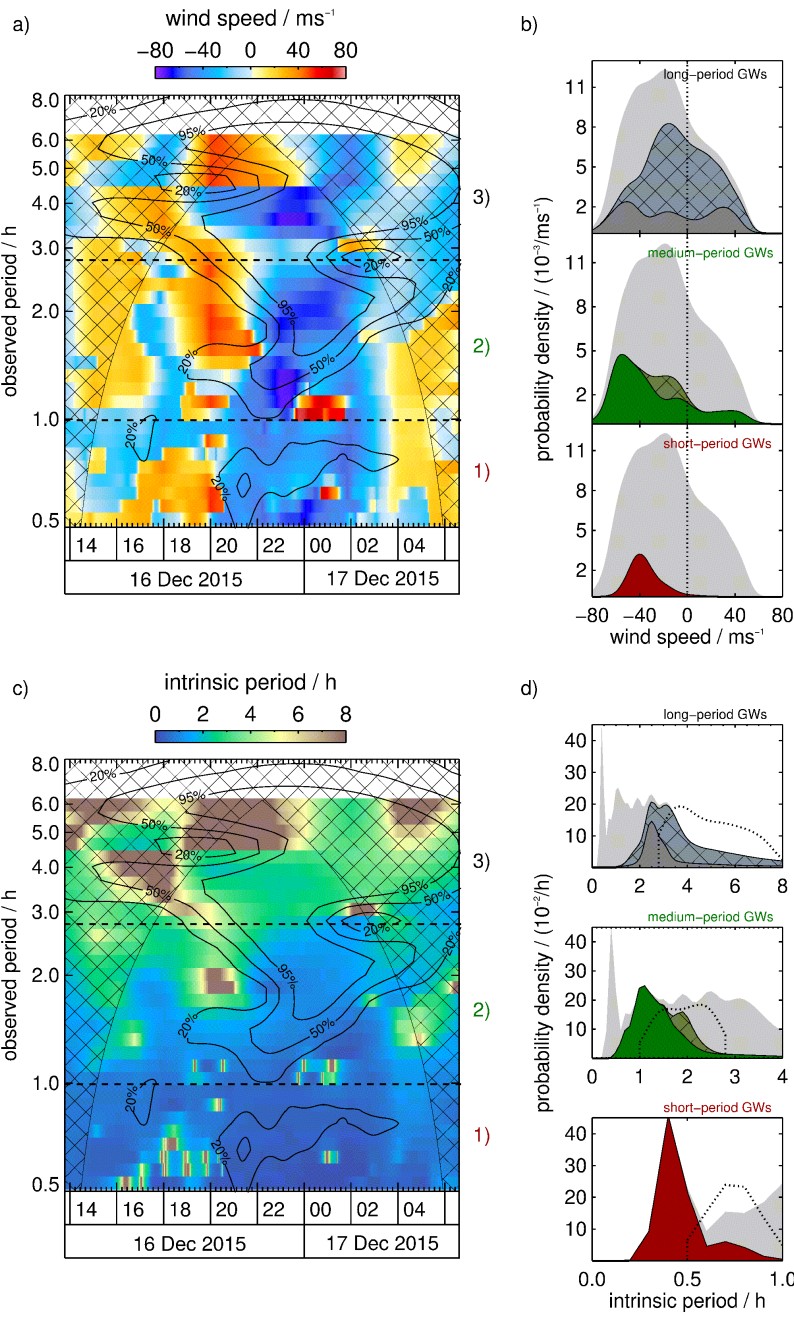

**Figure 8.** (a) Same as Fig. 7a for horizontal wind speeds in the direction of horizontal propagation. (b) Density distributions for each region as in Fig. 7b. The dotted line represents zero wind speed. (c) Same as Fig. 7a for intrinsic periods based on horizontal wavelengths and the background wind. (d) Density distributions for each region as in Fig. 7b. Additionally, the distributions of observed periods are given (dashed lines).



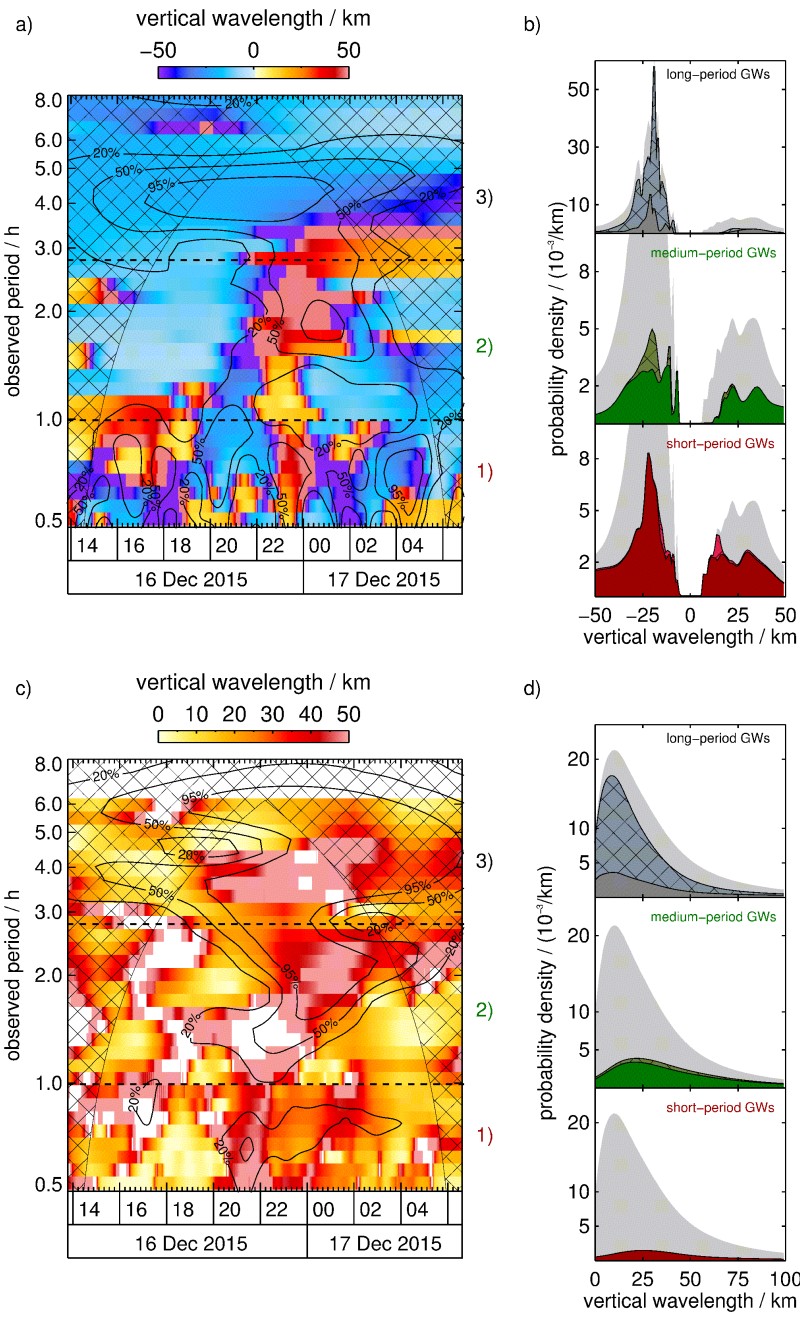

**Figure 9.** (a) Same as Fig. 7a for vertical wavelengths derived from lidar data. Positive vertical wavelengths indicate downward propagating waves (upward-slanted phase lines), negative vertical wavelengths indicate upward propagating waves (downward-slanted phase lines). (b) Density distributions for each region as in Fig. 7b. (c) Same as Fig. 7a for vertical wavelengths retrieved from the dispersion relation. White gaps indicate a complex wavelength. (d) Density distributions for each region as in Fig. 7b.





| Region | $\tau$ / h | time / UT | $\theta$ / ° | $\lambda_h$ / km | $\tau_I$ / h | $c_I$ / m/s | $u_0$ / m/s | $\lambda_z$ / km |
|---|---|---|---|---|---|---|---|---|
| 1) | 0.5 – 0.8 | 20 – 22 | 312 – 336 | | | | | |
| | 0.6 – 0.8 | 22 – 04 | 19 – 38 | 114 – 166 | 0.4 – 0.5 | 70 – 121 | -48 – -26 | 7 – 50 |
| | 0.8 – 1.0 | 17 | 125 – 162 | | | | | |
| 2) | 1.2 – 2.8 | 20 - 02 | 358 – 35 | 443 – 747 | 1.1 – 1.8 | 72 – 150 | -65 – -28 | 5 – 54 |
| | 1.2 – 1.6 | 19 – 22 | 88 – 177 | | | | | |
| 3) | 3.5 – 4.5 | 18 – 02 | 284 – 289 | | | | | |
| | | | 299 – 319 | | | | | |
| | 4.5 – 6.0 | 20 – 01 | 71 – 88 | 739 – 1032 | 2.5 – 3.1 | 21 – 113 | -65 – 43 | 1 – 32 |
| | 2.8 – 3.5 | 18 – 01 | 7 – 26 | | | | | |
| | | | 33 – 50 | | | | | |

Table 1. GW parameters derived from AMTM and SLICE data for selected dominant GW packets. The ranges describe the FWHM of peaks in each density distribution. Values for $\tau$ and time refer to identified peaks in the propagation direction.

| Region | $\tau$ / h | time / UT | $\lambda_z$ / km | propagation direction |
|---|---|---|---|---|
| 1) | 0.5 – 1.0 | whole night | 17 – 25 | upward |
| | | | 10 – 43 | downward |
| 2) | 1.0 – 1.5 | 20 – 22 | 11 – 36 | upward |
| | 1.0 – 2.0 | 02 – 06 | | upward |
| | 2.5 | 02 – 03 | 6 – 8 | upward |
| | 1.2 – 2.8 | 22 – 02 | 16 – 45 | downward |
| 3) | 3.5 – 6.0 | 18 – 02 | 19 – 22 | upward |
| | 2.8 – 3.5 | 18 – 22 | 12 – 14 | upward |
| | 2.8 – 3.5 | 22 – 03 | 24 – 40 | downward |

Table 2. Vertical wavelengths derived from CORAL data. The ranges describe the FWHM of peaks in the density distribution.

Table 1 and 2 summarize the range of GW parameters of a number of likely GW packets across the three spectral ranges for the discussion.

## 5 Discussion

5    Before we discuss the presented results of our analysis we want to point out the assumptions and limitations in this case study. As we mentioned in Sect. 2, the OH layer peak altitude is on average located at 86.6 km and has a thickness of 8.6 km. Because





no satellite soundings of OH are available for the period of the case study, we use these climatological values. However, we note that the actual peak altitude and thickness of the OH layer may be deviating from the climatological mean, leading to uncertainties in the wavelet spectra as we average Gaussian-weighted lidar temperatures over the altitude range of the OH layer. The same holds true for averaged wind data. Another limitation is that lidar measurements are restricted to the center of the OH imagers FOV with decreasing resolution towards higher altitudes. We highlighted the congruent altitude range of the three instruments data. However, resolutions and observational volumes differ. We chose a 20% significance level to maximize the overlapping areas in lidar and AMTM wavelet spectra especially for short periods, as Fig. 5c has proven that also at this significance levels detectable and coherent structures can be retrieved.

Dörnbrack et al. (2017) showed that upward propagating waves appear downward propagating in lidar data for $u_0 < -c_I$. In the majority of cases this condition is not fulfilled and the intrinsic vertical propagation direction matches the observed vertical propagation direction (upward/downward).

## 5.1 Short-period gravity waves

Spectral region 1) comprises GWs with observed periods between $0.5 - 1.0$ h. Comparison of the global wavelet spectra of CORAL (Fig. 4b) and the AMTM (Fig. 4e) yields coinciding peaks at $\sim 0.7$ h observed period. The difference in spectral amplitudes mentioned above is reflected in the levels of significance. The spectra in Fig. 4c show an increase of spectral amplitude with altitude. As long as no wave breaking occurs, we expect increasing amplitudes with altitude due to decreasing density. From comparing both spectra, we are confident to see the same GWs with both instruments but with different sensitivities at different times.

We find that the majority of horizontal wavelengths lies between $114 - 166$ km (Fig. 7b). When we compare the distribution of region 1) with the total density distribution (grey background), we discover that almost all GWs with smaller horizontal wavelengths ($<300$ km) are located within region 1). The density distributions reveal three preferred propagation directions (Fig. 7d). The FWHM of the three peaks are given in Table 1. We treat each peak as a GW packet. A first packet propagates southeastward at about 17 UT and a second one travels northwestward at 20 UT. The latter either turns northeastward after 22 UT, or a third packet appears. The background wind in propagation direction is negative at all times for all waves (Fig. 8b), indicating wave propagation against the mean flow. In accordance with the propagation against the background wind, intrinsic periods are smaller than observed periods (Fig. 8b). Again, almost all short-period GWs with intrinsic periods below 0.6 h are found in region 1). Our WAPITI algorithm identifies mainly upward propagating waves in lidar data with a majority of vertical wavelengths between $17 - 25$ km (Fig. 9b). The probability density distribution shows another broader peak between $10 - 43$ km affiliated with downward propagating waves. Figure 9ab also show vertical wavelengths larger than 50 km. We emphasize that these large values are retrieved only in the altitude range of the OH layer ($82.5 - 91.1$ km). Phase lines that are locally very steep e.g. due to vertical wind shear seem to exhibit very large vertical wavelengths. Upward and downward propagating waves are alternating throughout the night. Reason for this may be refraction levels or vertical wind shear. The calculated vertical wavelengths in Fig. 9d peak between $7 - 50$ km. The peaks identified in lidar data overlap largely with this





range.

In summary, region 1) contains short-period waves travelling predominantly northwestward against the background wind resulting in a Doppler-shift of intrinsic periods to larger observed periods. These short-period waves dominate the density distributions for horizontal wavelengths <300 km and intrinsic periods <0.6 h. In AMTM time lapse movies we observe GW breaking at 21 UT. This coincides with the appearance of large amplitudes in spectral region 1) in the AMTM wavelet spectrum. One explanation could be that short-period GWs are generated by longer-period GWs breaking at 21 UT. Amplitudes in the CORAL spectrum confirm the presence of long-period waves. Another possibility is a change of background conditions at 21 UT in such a way that afterwards short-period GWs can be detected by the AMTM. The rapid change of sign of the vertical wavelength shows that this region is dominated by vertical wind shear, i.e. $\frac{\partial u}{\partial z} \neq 0$. Alternating $\lambda_z$ may also be a sign for reflection of waves (ducted waves). The comparison of the two retrievals for vertical wavelengths shows that both methods are in good agreement. However, vertical wavelengths retrieved by the WAPITI algorithm have a narrower distribution than the values we derived from the dispersion relation. Discrete peaks in the density distribution of the propagation direction help to distinguish between GW packets.

## 5.2 Medium-period gravity waves

Region 2) comprises observed periods between 1.0 – 2.8 h. The peaks in the global wavelet spectra of CORAL (Fig. 4b) and the AMTM (Fig. 4e) are slightly shifted. For CORAL, the peak lies at 1.8 h while it is at 2.2 h for the AMTM. This difference may result from the different sensitivities of the instruments or the variability of the OH layer thickness and altitude. In comparison to region 1) we find a large overlapping area of statistical significance in both spectra. Figure 4c shows a slight increase of spectral amplitude with altitude compared to region 1) suggesting growing wave amplitudes.

Most of the horizontal wavelengths lie between 443 – 747 km (Fig. 7b). Taking the values within the COI into account does not alter in general the shape of the peak. When we compare the distribution of region 2) with the total density distribution (grey background) we see that region 2) contributes to all wavelengths. However, its contribution to larger wavelengths is weaker. Sometimes phase lines are locally very steep in both keograms and therefore $\lambda_h$ attains large values. One example is evident at 22 UT for 1.5 h observed period. After 23 UT, $\lambda_h$ decreases rapidly within one hour from 2000 km to 200 km. This is a clear sign for a gradient in the horizontal wind field as $\frac{\partial \lambda_h}{\partial t} \sim \frac{\partial u}{\partial x}$ (Marks and Eckermann, 1995; Stober et al., 2018). As evident from Fig. 7c the propagation direction is first westward and becomes northeastward later. Our explanation is that the waves appear to be rotating in AMTM data due to a horizontal wind gradient. This leads to bent phase lines in the temperature reconstructions which are identified as very large horizontal wavelengths. The dominant propagation direction is north- to northeastward. Later, waves turn westward. The northward propagation results in negative wind speeds (Fig. 8b). Only small spectral areas comprising waves with an eastward component experience positive wind speeds. Wind speeds from region 2) dominate the total density distribution (grey background) between -80 and -50 m/s. As we have seen in region 1), a propagation against the background wind leads to a Doppler-shift of intrinsic periods towards larger observed periods. This is in general also the case in region 2) where the majority of waves with intrinsic periods between 1.1 – 1.8 h is Doppler-shifted towards observed periods

between $1.0 - 2.8\,\mathrm{h}$ (Fig. 8c). Values within the COI slightly broaden the distribution. Intrinsic periods from region 2) dominate the total density distribution (grey background) between $0.6 - 2.1\,\mathrm{h}$. Most of the vertical wavelengths derived from lidar data lie between $11 - 36\,\mathrm{km}$ (Fig. 9b) and are affiliated with upward propagating waves. We identify two additional peaks between $6 - 8\,\mathrm{km}$ (upward) and $16 - 45\,\mathrm{km}$ (downward). On the left of Fig. 9a we find waves first propagating upward, then turning

downward at 22 UT and back upward at $\sim$02 UT. As mentioned in region 1) the change of propagation direction indicates vertical wind shear or levels of reflection (ducted waves). The distribution of vertical wavelengths estimated using the dispersion relation shows an even broader peak as in region 1) with values between $5 - 54\,\mathrm{km}$ (Fig. 9b). Vertical wavelengths within the COI increase slightly the probability density at the position of the peak. In general both density distributions of vertical wavelengths (Fig. 9bd) are in good agreement. Eye-catching is the white gap between $21 - 23$ UT in Fig. 9c. In this area the

condition for a real $\lambda_z$ is not fulfilled. Waves in this area are either not propagating or the vertical wind shear is underestimated.

We state that the appearance of GWs in region 2) coincides with a wave breaking event at 21 UT and a strengthening of the mean flow. The retrieved parameters show a large variability which may follow from a non-uniform wind speed distribution within the FOV of the AMTM and altitude range of the OH layer. Interestingly not all parts of the GW spectrum react in the

same way to the changing background wind.

### 5.3 Long-period gravity waves

A wide range of observed periods from $2.8 - 6.5\,\mathrm{h}$ is comprised in spectral region 3). When we compare the global wavelet spectra of CORAL (Fig. 4b) and the AMTM (Fig. 4e) we find the most prominent peak at $4.2\,\mathrm{h}$ observed period in the CORAL spectrum and a slightly shifted peak at $3.8\,\mathrm{h}$ in the AMTM spectrum. We assert that both instruments are sensitive to waves

in this spectral domain, although, some exceptions are striking. The contour line of the 20% significance level in the AMTM spectrum covers the whole night and comprises observed periods between $2.8 - 8.0\,\mathrm{h}$. The statistically significant area in the CORAL spectrum is more focused and comprises observed periods between $2.8 - 6.0\,\mathrm{h}$. Right at the position of maximum amplitude in the CORAL spectrum we find a gap in the spectrum of the AMTM. This gap results most probably from waves with small vertical wavelengths which cannot be detected by the AMTM. Another explanation is a higher/lower than assumed

OH layer altitude. Looking at Fig. 4c we find a peak shifting from $3.5\,\mathrm{h}$ observed period at 82 km altitude to $4.5\,\mathrm{h}$ at 90 km. This behaviour might be indicative of separate waves at different altitudes, or the same wave being Doppler-shifted to multiple observed periods at different altitudes, which then implies a vertical gradient in horizontal wind speed. Figure 3 adumbrates such a vertical gradient. From $\frac{\partial \lambda_z}{\partial t} \sim \frac{\partial u}{\partial z}$ we expect a changing vertical wavelength with time as we find a vertical gradient in horizontal wind speed. Figures 9ac show a changing $\lambda_z$.

We find a rather broad distribution of horizontal wavelengths with a peak between $739 - 1032\,\mathrm{km}$ (Fig. 7b). Eye-catching are two areas reaching at least 2000 km horizontal wavelength. A major peak is located at 01 UT and $4.4\,\mathrm{h}$ observed period and a smaller peak at 0130 UT and $3.3\,\mathrm{h}$ observed period. As mentioned above, phase lines appear locally steep in the temperature reconstructions likely due to a gradient in horizontal wind. At the same time, waves appear to be rotating. Both peaks in Fig. 7a show a rapid turning of waves with an angular frequency of $> 90°$ within 2 h. Overall, considering the distribution of



propagation directions we identify three different GW packets (Fig. 7d). The first comprises observed periods between 2.8 – 3.5 h and propagates in a north- northeastward direction between 18 – 01 UT. The second packet exhibits observed periods between 3.5 – 4.5 h, propagates westward and turns northward during the night. Between 4.5 – 6.0 h observed period we find a third wave packet propagating eastward and turning northward during the duration of the measurement. Interesting to observe

is the bidirectionality of the GW packets which is also reflected in the distribution of wind speeds. Most of the waves propagate against the wind, i.e. wind speeds are negative, but there is also a large part of waves travelling with the wind (Fig. 8b). The first GW packet experiences tailwind at 18 UT but at 19 UT the wind turns and strengthens to negative values. Negative winds are present for the second GW packet between 20 – 04 UT as well. The third packet experiences positive and decreasing wind speeds between 18 – 01 UT. Values inside the COI modify the distribution such that it changes from a plateau to a peak shape.

Wind speeds from region 3) dominate the total density distribution (grey background) between -50 and 80 m/s. Figure 8b shows that most of the waves outside the COI are Doppler-shifted from small intrinsic periods between 2.5 – 3.1 h to larger observed periods between 2.8 – 6.5 h. Intrinsic periods from region 3) dominate the total density distribution (grey background) between 2.1 – 8.0 h. Based on lidar data the WAPITI algorithm identifies a majority of vertical wavelengths between 19 – 22 km (Fig. 9a). Vertical wavelengths from region 3) dominate the total density distribution (grey background) for negative values. We

find two smaller peaks between 12 – 14 km (upward) and 24 – 40 km (downward). This is in good agreement with values retrieved using the dispersion relation (Fig. 9b). Due to large uncertainties in the calculation of vertical wavelengths the density distribution is very broad. Hence, values for $\lambda_z$ smaller than the OH layer thickness are also retrieved. Looking to the left of Fig. 9c we find another white gap indicating a complex vertical wavenumber. This area coincides with a domain of extremely large horizontal wavelengths (Fig. 7a) and fast wave rotation (Fig. 7c). As mentioned above this is an indication for a gradient

in the horizontal wind field. Vertical wavelengths are relatively small and slowly increasing, which is in agreement with the non-detection by the AMTM in the beginning of the measurement due to the lack of sensitivity to small vertical wavelengths. At the same time the variation of $\lambda_z$ gives evidence for a vertical wind shear. We rearrange the ray tracing equations stated by Marks and Eckermann (1995) yielding

$$\frac{\partial u}{\partial z} = \frac{\lambda_h}{\lambda_z^2} \frac{d\lambda_z}{dt} \tag{12}$$

with $u$ the horizontal wind speed in an arbitrary direction. $\lambda_h$ and $\lambda_z$ are averaged at an observed period of 4 h between 20 – 03 UT, i.e. a time span in which both wavelet spectra exhibit large significance levels. The vertical wavelength changes from -15 km to -30 km within this time span. Hence, $\frac{\partial u}{\partial z} = \frac{900\,\mathrm{km}}{20^2\,\mathrm{km}^2} \frac{15\,\mathrm{km}}{7\,\mathrm{h}} = 1.3 \frac{\mathrm{m\,s^{-1}}}{\mathrm{km}}$. When we multiply this value with the thickness of the OH layer we get a total difference in wind speed of $\Delta u = 11.5\,\mathrm{m\,s^{-1}}$. The mean absolute wind speed retrieved from SLICE between 20 – 03 UT increases from 41 $\mathrm{m\,s^{-1}}$ at 82 km to 51 $m\,s^{-1}$ at 90 km (Fig. 3). This value is in very good

agreement with our estimate. We conclude that with the help of our method we are able to derive vertical wind gradients and possibly even horizontal gradients.

In region 3) we identify two GW packets moving in opposite directions (west- and eastward) turning slowly northward. Both propagate upward. These changes of propagation direction are most probably due to wind gradients. The existence of





long-period waves is one condition for the supposed gravity wave breaking at 21 UT.

In summary, we demonstrated that our analysis is capable of determining the full set of GW parameters covering a wide range of values that are known to be typical for gravity waves. Typical values for $\lambda_h$ in airglow data are in the range of

5   $10 - 200\,\mathrm{km}$ (Nakamura et al., 2003; Diettrich et al., 2005; Matsuda et al., 2014; Lu et al., 2015; Nyassor et al., 2018). By analysing keograms, Fritts et al. (2014) retrieve even larger $\lambda_h$ with values reaching $4000\,\mathrm{km}$. They decompose temperature perturbations into few dominant modes and find a wide range of values for horizontal wavelength ($24 - {\sim}4000\,\mathrm{km}$), vertical wavelength ($17.6 - {\sim}30\,\mathrm{km}$), intrinsic period (${\sim}10\,\mathrm{min} - {\sim}12\,\mathrm{h}$), intrinsic phase velocity ($33 - {>}200\,\mathrm{m\,s^{-1}}$) and background wind (${\sim}14 - 51\,\mathrm{m\,s^{-1}}$). Vertical wavelengths retrieved from lidar data are typically between $2 - 20\,\mathrm{km}$ (Kaifler et al., 2017).

10   Values for $\lambda_z > 50\,\mathrm{km}$ are an indication for ducted waves (Snively and Pasko, 2003).





# 6 Conclusions

In this study we combined three complementary data sets obtained from co-located instruments. Vertical temperature profiles by the CORAL lidar and the AMTM's horizontal temperature maps provide three-dimensional insight in the behaviour of GWs in the MLT region. Additional wind information provided by the SLICE meteor radar made it possible to investigate the

intrinsic propagation of these GWs. Our newly developed WAPITI algorithm combines spectral filtering using wavelet analysis with a phase line identification algorithm. Based on this method, we were able to retrieve observed as well as intrinsic GW parameters with estimations of their uncertainties as a function of time and ground-relative period. This facilitates separation and characterization of GW packets without using the dispersion relation.

Although the sensitivities of the instruments differ, by comparing wavelet spectra we confirm that AMTM and CORAL ob-

served in general the same GWs. For the case study on 16/17 December 2015, the night started with large scale waves between $3-5\,\mathrm{h}$ ground-relative period. At 21 UT wave breaking occurred resulting in a spectral broadening and creation of short-period waves. The mean flow turns into a southeastward direction and strengthens. The detected GWs propagate predominantly against this background wind in a northward direction resulting in a Doppler-shift of about $1\,\mathrm{h}$. The vertical wind shear caused a steepening of phase lines, i.e. an increase of vertical wavelengths. Additionally we find very large horizontal wavelengths

and wave rotation indicating a horizontal wind shear as well. We were not looking for isolated wave events but investigated the whole data sets, interpreting the observations as a superposition of several GWs. All retrieved parameters are highly variable in time and observed period. This is an evidence for a non-uniform wind field in space and time and points out the complex interaction between waves and the background flow. As only parts of the spectra are sensitive to wind gradients, we conclude that short-period waves see long-period ones as disturbances in the background. Only the distribution of propagation directions

exhibited multiple discrete peaks which help to distinguish clearly between GW packets. The GW parameters can be used to calculate momentum fluxes, to perform forward and backward ray tracing and to derive a horizontally resolved wind field. The largest uncertainty of intrinsic parameters derived with our method arises due to the unknown precise altitude and shape of the OH layer. A reliable method to determine the precise OH profile is needed in order to improve GW results and to allow for better differentiation between GW packets. We plan to automatise our method and apply it to more case studies and eventually

to the whole data set with the goal to study propagation and interaction of GWs with the mean flow from a statistical point of view. In order to assess spatial properties of GWs we want to extend our analysis to the whole FOV of the imager.

*Code and data availability.* Lidar, radar and AMTM datasets are available as netcdf files in the HALO-DB at https://halo-db.pa.op.dlr.de/mission/109. Entries 6457 to 6469.

# Appendix A: Uncertainties

In this section we present uncertainty calculations for the retrieved parameters horizontal wavelength, direction of propagation, intrinsic period, wind in propagation direction, wind shear, wind curvature and vertical wavelength derived from the disper-





sion relation. The uncertainty of the intrinsic period contains wind speed uncertainties $\Delta u_0$ which comprises three sources. First we average the Gaussian weighted wind speed over an altitude range $82 - 95\,\mathrm{km}$ with a FWHM of $8.6\,\mathrm{km}$ centered at $86.8\,\mathrm{km}$. This is the average altitude and thickness of the OH layer. As mentioned in Sect. 2.2, these parameters are variable, therefore it is possible that the OH layer is partly below $82\,\mathrm{km}$ or above $91\,\mathrm{km}$ and we are not averaging over the correct

altitude range. Second, we calculate the wind speed in the direction of propagation which is based on the estimated $\theta$. As $\theta$ may be inaccurate, there is some uncertainty in the projected wind speed as well. Finally, for the wind measurements itself we assume uncertainties of $10\,\mathrm{m\,s^{-1}}$. As we average over at least three independent values, the uncertainty is reduced to $5.6\,\mathrm{m\,s^{-1}}$.

$$\Delta\lambda_x = \tau\Delta c_x \tag{A1}$$

$$\Delta\lambda_y = \tau\Delta c_y \tag{A2}$$

$$\Delta\lambda_h = \left|\frac{\partial\lambda_h}{\partial\lambda_x}\right|\Delta\lambda_x + \left|\frac{\partial\lambda_h}{\partial\lambda_y}\right|\Delta\lambda_y \tag{A3}$$

$$= \frac{|\lambda_y|\Delta\lambda_x + |\lambda_x|\Delta\lambda_y}{(\lambda_x^2 + \lambda_y^2)^{3/2}}. \tag{A4}$$

$$\Delta\theta = \frac{|\lambda_y|\Delta\lambda_x + |\lambda_x|\Delta\lambda_y}{\lambda_y^2 + \lambda_x^2}. \tag{A5}$$

$$\Delta\tau_I = \left|\frac{\partial\tau_I}{\partial\lambda_h}\right|\Delta\lambda_h + \left|\frac{\partial\tau_I}{\partial u_0}\right|\Delta u_0 \tag{A6}$$

$$= \frac{|u_0|\Delta\lambda_h + |\lambda_h|\Delta u_0}{(c - u_0)^2}. \tag{A7}$$

$$\Delta u_0(\theta) = \left|\frac{\partial u_0}{\partial\theta}\right|\Delta\theta + \left|\frac{\partial u_0}{\partial U}\right|\Delta U + \left|\frac{\partial u_0}{\partial V}\right|\Delta V \tag{A8}$$

$$= |U\cos(\theta) - V\sin(\theta)|\Delta\theta + |sin(\theta)|\Delta U + |cos(\theta)|\Delta V. \tag{A9}$$

$$\Delta u_0' = \frac{\sqrt{2}\Delta u_0}{dz} \tag{A10}$$

$$\Delta u_0'' = \frac{2\Delta u_0}{dz^2} \tag{A11}$$



$$\Delta\lambda_z = \left|\frac{\partial\lambda_z}{\partial\lambda_h}\right|\Delta\lambda_h + \left|\frac{\partial\lambda_z}{\partial\tau_I}\right|\Delta\tau_I + \left|\frac{\partial\lambda_z}{\partial N}\right|\Delta N + \left|\frac{\partial\lambda_z}{\partial H_s}\right|\Delta H_s + \left|\frac{\partial\lambda_z}{\partial u_0'}\right|\Delta u_0' + \left|\frac{\partial\lambda_z}{\partial u_0''}\right|\Delta u_0'' \tag{A12}$$

$$= \pi\left(\frac{\lambda_z}{2\pi}\right)^3 \left|\frac{8\pi^2 - 2N^2\tau_I^2}{\lambda_h} + \frac{u_0'\tau_I}{H_s} - u_0''\tau_I\right|\frac{\Delta\lambda_h}{\lambda_h^2} \tag{A13}$$

$$+ \pi\left(\frac{\lambda_z}{2\pi}\right)^3 \left|\frac{2N^2\tau_I}{\lambda_h} - \frac{u_0'}{H_s} + u_0''\right|\frac{\Delta\tau_I}{|\lambda_h|} \tag{A14}$$

$$+ \pi\left(\frac{\lambda_z}{2\pi}\right)^3 \left|\frac{2N\tau_I^2}{\lambda_h^2}\right|\Delta N \tag{A15}$$

$$+ \pi\left(\frac{\lambda_z}{2\pi}\right)^3 \left|\frac{1}{H_s} + \frac{u_0'\tau_I}{\lambda_h}\right|\frac{\Delta H_s}{H_s^2} \tag{A16}$$

$$+ \pi\left(\frac{\lambda_z}{2\pi}\right)^3 \left|\frac{\tau_I}{H_s\lambda_h}\right|\Delta u_0' \tag{A17}$$

$$+ \pi\left(\frac{\lambda_z}{2\pi}\right)^3 \left|\frac{\tau_I}{\lambda_h}\right|\Delta u_0''. \tag{A18}$$

**A1**

*Author contributions.* R. Reichert developed the method, carried out all data analysis and wrote the majority of the manuscript. The idea was suggested by B. Kaifler who also supervised the work. N. Kaifler provided the lidar data, wrote part of the introduction and revised the manuscript. M. Rapp supervises the doctoral thesis of R. Reichert and made suggestions. P.-D. Pautet and M. Taylor provided AMTM data. A. Kozlovsky and M. Lester provided wind data. R. Kivi supported the lidar and AMTM operation at Sodankylä.

*Competing interests.* The authors declare to have no competing interests.

*Acknowledgements.* R. Reichert thanks the German Research Foundation (DFG) for support through the research unit Multiscale Dynamics of Gravity Waves (MS-GWaves) grant RA 1400/6-1. This work was partly supported by the ARISE2 project (http://arise-project.eu/) within the HORIZON 2020 program of the European Commission. N. Kaifler was supported in the development of CORAL by the Helmholtz association within the project PD-206. The development and operations of AMTMs were funded by the AF DURIP grant F49620-02-1-0258, and the NSF grants AGS-1061892 and AGS-1042227. The authors thank the personnel of the Finnish Meteorological Institute in Sodankylä who provided assistance with the installation and operations of the CORAL lidar and AMTM imager.





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
