# Peer review of "Retrieval of intrinsic mesospheric gravity wave parameters using lidar and airglow temperature and meteor radar wind data"

_Atmospheric Measurement Techniques, 2019_

## Referee Comment (RC1) · Anonymous Referee #2 · 7 Jun 2019

Comments on 'Retrieval of intrinsic mesospheric gravity wave parameters using lidar and airglow temperature and meteor radar wind data' by R. Reichert et al

This paper describes a method to combine temperature observations made by a lidar (in the vertical) and a temperature mapper (horizontally over the hydroxyl layer) to obtain gravity-wave parameters. The method enhances its outputs by including meteor-radar winds which allow for the extraction of some intrinsic parameters. The method is applied to two days of observations and the results are interpreted.

The paper as presented is well thought out, communicates its ideas well and demonstrates a strong knowledge of the observing techniques and the gravity waves it seeks

to understand. Some global comments made below may improve the manuscript and some grammatical notes are included.

It is notable that there is no mention of tides in the paper, despite them having a major influence on the background environment in the height regions being considered. It is reasonable to consider the tides as simply a background environment for the gravity waves but they should still be mentioned in the context of the work.

One of the challenges that the authors have to meet is the observation of different wave characteristics using the two instruments at similar places and times. They discuss the limited knowledge of the OH layer height and profile on P23 and note in the conclusions that improvements in this area are needed. Temporal and spatial variations in the wind field are also noted as strong influences on the measurement and evolution of the gravity waves characteristics. I have a concern that the wind speeds obtained by the meteor radar, which use all-sky detections and thus average over a large area of the sky, may also introduce some discrepancies. Consideration of this should be included in the discussion.

The authors do a good job of aligning the instrumental filter functions of the instruments (using the OH layer weighting) so that the sensitivity of the instruments to the GW spectrum is matched. This a strength they should note explicitly in the context of the work of Alexander (1998). (Re P23 L23.)

Technical and grammatical comments:

I think the inclusion of a diagram showing the form of the wavelet function would improve section 3.1

P8 L22 Delete 'times'

P8 L24 change 'standard deviation is shown as dashed line' to 'one standard deviation range is shown with dashed lines'

P9 L2 'If GWs with periods tau_j are present in the data set'... What criterion are you

using to say a GW is present?

Fig 5: the 'a)' label is too large.

P11 L7 Are OH layer weightings included in the weightings for the linear fits against height? If not, should they be?

P13 L3 insert 'projected onto the wind direction' after 'background wind'

P13 L 10 Suggest replace 'uniform' with 'no horizontal gradients in the'

P13 L13 – Insert a capital omega after 'intrinsic frequency'. Personally I would prefer you used the nomenclature in Fritts and Alexander (2003) and had an omega hat here. The u_o would also be changed.

P15 L7 'Figure 4b' should be 'Figure 4d'

P15 L8 'Figure 4bcd' should be 'Figure 4bce'

P23 L16 suggest change 'comprised' to 'included'

P23 L27 Thank you for introducing me to the word 'adumbrates' but I am afraid it is not a good fit here. It should be replaced by 'supports the presence of'

P23 L32 Suggest insert 'in propagation direction' after 'rotating'.

P23 L33 replace 'frequency' with 'change'

P25 L7 insert 'a' before 'few'

P26 L17 Change 'This is an' to 'This provides'

P26 L24 change 'automatise' to 'automate'

---

## Referee Comment (RC2) · Anonymous Referee #1 · 28 Jun 2019

This is a nice study presenting lidar and IR temperature measurements, and meteor radar wind measurements in the mesosphere. The authors presenting a method to combine the data to provide physically useful insight in the gravity wave structures. The authors do a nice job describing their method, which, as they say, combines spectral filtering using wavelet analysis with a phase line identification algorithm. The clear physical descriptions of exactly what each instrument is actually measuring help to ensure the reader that the authors understand not only the analysis, but also the measurements. I have only a few minor suggestions for improvements to the manuscript.

Page 2 line 9-10: "Limitations . . ." This is an awkward sentence and I'm not quite sure

what point is being made.

Page 3 line 1 – Several abbreviations are given here, but they are not fully spelled out until page 4.

Page 3 line 7 – should say "deriving" and "studying"

Page 20 line 31 – The use of e.g. in this way is a bit awkward. Perhaps it would be better to place the phrase "e.g. due to vertical wind shear" inside parentheses.

Figure 9, and discussion in 5.1 and 5.2 – In both 5.1 and 5.2 there is the statement that 9b and 9d "are in good agreement". Please provide some quantification of what is meant by this. Given the different scales and the fact that 9b has positive and negative wavelengths it is difficult to visually determine the level of agreement from the figure.

Page 22 line 17 – "comprised" is not the right word here.

Page 22 line 27 - I asked 4 fellow native English speakers what "adumbrates" and no one knew. Still, it seems appropriate, so it is okay to keep it here if you like.

Page 22 last line – "angular frequency" is certainly not the right phrase here.

Page 23 line 17 – "looking to the left of" should be replaced with an appropriate date/time range.

---

## Author Comment (AC1) · 20 Aug 2019

Author's final response

We thank the referee for the comments. Please find attached the latexdiff version. Page and Line references in the latexdiff version are given in brackets.

Anonymous Referee #1

This is a nice study presenting lidar and IR temperature measurements, and meteor radar wind measurements in the mesosphere. The authors presenting a method to

combine the data to provide physically useful insight in the gravity wave structures. The authors do a nice job describing their method, which, as they say, combines spectral filtering using wavelet analysis with a phase line identification algorithm. The clear physical descriptions of exactly what each instrument is actually measuring help to ensure the reader that the authors understand not only the analysis, but also the measurements. I have only a few minor suggestions for improvements to the manuscript.

P2 L9-10 - "Limitations..." This is an awkward sentence and I'm not quite sure what point is being made.

> We changed the sentence to: P2 L9-10 (P2 L9-11) "The detection of GW by means of OH layer intensity observations depends on the (usually unknown) width and height of the OH layer as well as the GW period and vertical wavelength (Gardner and Taylor, 1998; Dunker, 2018)."

P3 L1 – Several abbreviations are given here, but they are not fully spelled out until page 4.

> We spelled out the abbreviations on P3 L1-2 (P3 L3-4) and use instead abbreviations on P4 L2-3 (P4 L5-6).

P3 L7 – should say "deriving" and "studying"

> Corrected. P3 L8 (P3 L10)

P20 L31 – The use of e.g. in this way is a bit awkward. Perhaps it would be better to place the phrase "e.g. due to vertical wind shear" inside parentheses.

> We changed that. P22 L31 (P22 L33)

Figure 9, and discussion in 5.1 and 5.2 – In both 5.1 and 5.2 there is the statement that 9b and 9d "are in good agreement". Please provide some quantification of what is meant by this. Given the different scales and the fact that 9b has positive and negative wavelengths it is difficult to visually determine the level of agreement from the figure.

> We have added a reference to the comparison of FWHM values given in Table 1 and 2 as quantification. We decided to show the sign of the vertical wavelength in Fig. 9ab in order to evaluate the vertical propagation direction, but we have improved the text to make clear that only absolute values are compared. P24 L7-8 (P24 L10-11)

P22 L17 – "comprised" is not the right word here.

> We changed 'comprised' to 'detected. P24 L17 (P24 L20)

P22 L27 - I asked 4 fellow native English speakers what "adumbrates" and no one knew. Still, it seems appropriate, so it is okay to keep it here if you like.

> The sentence was changed to 'Such a vertical gradient is supported by SLICE meteor wind measurements (Fig. 3). P24 L27-28 (P24 L31)

P22 L34 – "angular frequency" is certainly not the right phrase here.

> We changed it to 'angular change' as suggested by the second reviewer. P25 L1 (P25 L4)

P23 L17 – "looking to the left of" should be replaced with an appropriate date/time range.

> We erased 'looking to the left of'. Intentionally we just wanted to guide the eye from the right Figure, i.e. Figure 9d, to the left Figure, i.e. Figure 9c, and not the left part of Figure 9c. P25 L18 (P25 L22)

Please also note the supplement to this comment:
https://www.atmos-meas-tech-discuss.net/amt-2019-73/amt-2019-73-AC1-supplement.pdf

———————————————

[Figure]

**Supplement:**

[revised manuscript text omitted]

---

## Author Comment (AC2) · 20 Aug 2019

Author's final response

We thank the referee for the comments. Please find attached the latexdiff version. Page and Line references in the latexdiff version are given in brackets

Anonymous Referee #2

Received and published: 7 June 2019

Comments on 'Retrieval of intrinsic mesospheric gravity wave parameters using lidar and airglow temperature and meteor radar wind data' by R. Reichert et al This paper Printer-friendly version

describes a method to combine temperature observations made by a lidar (in the vertical) and a temperature mapper (horizontally over the hydroxyl layer) to obtain gravitywave parameters. The method enhances its outputs by including meteor-radar winds which allow for the extraction of some intrinsic parameters. The method is applied to two days of observations and the results are interpreted. The paper as presented is well thought out, communicates its ideas well and demonstrates a strong knowledge of the observing techniques and the gravity waves it seeks to understand. Some global comments made below may improve the manuscript and some grammatical notes are included.

It is notable that there is no mention of tides in the paper, despite them having a major influence on the background environment in the height regions being considered. It is reasonable to consider the tides as simply a background environment for the gravity waves but they should still be mentioned in the context of the work.

> We have added the discussion of tides at several places within the manuscript: We emphasized on P16 L10-14 (P16 L12-15) that our method is insensitive to tides as we only consider periods smaller than 6.5 hours. Additionally we state that we treat tides as a background that may interact with GWs and alter their parameters. Additionally we mentioned tides on P23 L7-8 (P23 L9-10) in the context of changing background conditions. Finally we state an alternative reason on P26 L3-4 (P26 L6-7) for a GW breaking process.

One of the challenges that the authors have to meet is the observation of different wave characteristics using the two instruments at similar places and times. They discuss the limited knowledge of the OH layer height and profile on P23 and note in the conclusions that improvements in this area are needed. Temporal and spatial variations in the wind field are also noted as strong influences on the measurement and evolution of the gravity waves characteristics. I have a concern that the wind speeds obtained by the meteor radar, which use all-sky detections and thus average over a large area of the sky, may also introduce some discrepancies. Consideration of this should be included
in the discussion.

> We added two sentences on P22 L3-4 (P22 L5-6) stating that wind speeds are averaged over a circle of 300km in diameter and therefore uncertainties are large. Additionally we refer to the appendix.

The authors do a good job of aligning the instrumental filter functions of the instruments (using the OH layer weighting) so that the sensitivity of the instruments to the GW spectrum is matched. This is a strength they should note explicitly in the context of the work of Alexander (1998). (Re P23 L23.)

> The reference to Alexander (1998) was added on P8 L25-26. (P8 L27-28)

Technical and grammatical comments: I think the inclusion of a diagram showing the form of the wavelet function would improve section 3.1

> We think that a diagram showing the form of the wavelet function is unnecessary as the wavelet transformation is commonly applied to atmospheric data sets and the Morlet wavelet is the one most often used.

P9 L2 - 'If GWs with periods tau\_j are present in the data set' ... What criterion are you using to say a GW is present?

> This is decided on the amplitude of the reconstructed period (or spectral power, and respective significance levels). A high amplitude indicates a strong contribution from this part of the spectrum, whereas a vanishing amplitude indicates the absence of GW with this period. We have added "phase lines of significant amplitude" to the text. P9 L5 (P9 L6)

P11 L7 - Are OH layer weightings included in the weightings for the linear fits against height? If not, should they be?

> OH layer weightings are not included in the linear fit. A Gaussian weighting would increase uncertainties for retrieved vertical wavelengths in an unpredictable manner.
Besides we compared fitting results using the two weightings, boxcar and Gaussian, and found no significant difference.

P13 L3 - insert 'projected onto the wind direction' after 'background wind'

> The reviewer probably meant to say "projected onto the GW propagation direction", we changed that. P13 L12 (P13 L13)

P13 L13 – Insert a capital omega after 'intrinsic frequency'. Personally I would prefer you used the nomenclature in Fritts and Alexander (2003) and had an omega hat here. The u\_o would also be changed.

> We follow the nomenclature in An Introduction to Atmospheric Gravity Waves by C. Nappo.

> We have made all corrections as suggested below.

P8 L22 - Delete 'times'

P8 L24 - change 'standard deviation is shown as dashed line' to 'one standard deviation range is shown with dashed lines'

Fig 5 - the 'a)' label is too large.

- P13 L10 Suggest replace 'uniform' with 'no horizontal gradients in the'
- P15 L7 'Figure 4b' should be 'Figure 4d'
- P15 L8 'Figure 4bcd' should be 'Figure 4bce'
- P22 L17 suggest change 'comprised' to 'included'

P22 L27 - Thank you for introducing me to the word 'adumbrates' but I am afraid it is not a good fit here. It should be replaced by 'supports the presence of'

> The sentence was changed.

P22 L33 - Suggest insert 'in propagation direction' after 'rotating'.
P22 L34 - replace 'frequency' with 'change'

- P24 L7 insert 'a' before 'few'
- P25 L17 Change 'This is an' to 'This provides'
- P25 L24 change 'automatise' to 'automate

Please also note the supplement to this comment: https://www.atmos-meas-tech-discuss.net/amt-2019-73/amt-2019-73-AC2supplement.pdf

**AMTD**

**Supplement:**

[revised manuscript text omitted]